# Finite Element Study for Mass Sensitivity of Love Surface Acoustic Wave Sensor with Si_3_N_4_-SiO_2_ Double-Covered Waveguiding Layer

**DOI:** 10.3390/mi14091696

**Published:** 2023-08-30

**Authors:** Luming Li, Mingyong Zhou, Lei Huang, Bingyan Jiang

**Affiliations:** 1State Key Laboratory of Precision Manufacturing for Extreme Service Performance, Central South University, Changsha 410083, China; 2College of Mechanical and Electrical Engineering, Central South University, Changsha 410083, China

**Keywords:** L-SAW, double-covered waveguiding layer, mass sensitivity, finite element method (FEM)

## Abstract

Love surface acoustic wave (L-SAW) sensors are miniaturized, easy to integrate, and suitable for detection in liquid environments. In this paper, an L-SAW sensor with a thin Si_3_N_4_-SiO_2_ double-covered layer was proposed for samples with small mass loads. The output response, phase velocity of the acoustic wave, and the mass sensitivity were analyzed using the finite element method (FEM). The simulation results show that the Si_3_N_4_ layer with high wave velocity greatly weakens the limitation of SiO_2_ on the phase velocity. The phase velocity can reach about 4300 m/s, which can increase the frequency shift when the same mass load is applied. Within a certain range, the mass sensitivity of the sensor is enhanced with the increase in the total thickness of the waveguiding layer and the thickness ratio of Si_3_N_4_ in the double-covered layer. When the thickness ratio is 1:2, the peak value of the mass sensitivity of the sensor is approximately 50% higher than that achieved with only the SiO_2_ waveguiding layer. The surface average stress of the delay line region follows the same trend as the mass sensitivity. The increase in mass sensitivity is the result of the heightened stress on the sensor surface. This L-SAW sensor, featuring a double-covered waveguiding layer, demonstrates high sensitivity and a simple structure. The simulation results lay a foundation for the design and manufacture of SAW sensors.

## 1. Introduction

Surface acoustic wave (SAW) sensors are widely used in the detection of mass load, strain, pressure, temperature, and acceleration [1,2,3,4]. They are favored for their simple structure, label-free operation, low cost, and easy integration [5,6]. Based on their structure [7], SAW sensors can be categorized as delay line type, resonant type, oscillation type, etc. Delay line sensors have a straightforward structure, enabling easy signal extraction. Their large sensing region is advantageous for forming functionalized membranes and preventing short circuits of interdigital electrodes when used in a liquid environment.

SAW can be divided into Rayleigh surface acoustic wave (R-SAW) and shear-horizontal surface acoustic wave (SH-SAW) based on the vibration direction of the particles relative to the device’s surface. R-SAW is suitable for detecting gas samples as its particle vibration is perpendicular to the device’s surface. However, when used in a liquid environment, the acoustic wave energy is significantly attenuated due to radiation into the liquid. On the other hand, SH-SAW-based sensors are widely used for liquid detection as particles vibrate parallel to the surface, preventing energy leakage into the liquid. Kwon et al. [8] successfully fabricated a delay line SH-SAW biosensor using a 36°Y-X LiTaO_3_ piezoelectric substrate with a gold nanolayer as a sensitive layer, achieving quantitative detection of HigG molecules. Sensitivity is a critical indicator of SAW sensors and significantly impacts their detection performance. Researchers have conducted relevant studies on optimizing sensor structures to enhance SH-SAW sensor sensitivity. Trivedi et al. [9] constructed an SH-SAW sensor containing the S1813 micro-ridge to achieve quantitative detection of avidin. Through simulations and experiments, it was found that the mass sensitivity of the device becomes maximum when coupled with resonance, and the average stress of the area at the interface between the ridge and the substrate increases. Li et al. [10] used 64°-YX LiNbO_3_ as the piezoelectric substrate and a series of trench structures were designed on the delay line path. The finite element method (FEM) results indicated that the presence of a trench increases the surface stress of the sensor, effectively improving its sensitivity. Although the SH-SAW sensor is suitable for detection in liquid environments, its energy gradually leaks into the substrate as acoustic waves propagate, which will result in a lower utilization rate of acoustic energy. Moreover, the liquid environment can cause corrosion of the electrodes on the surface of the piezoelectric substrate.

Similar to the Rayleigh wave, an SH wave with a decaying amplitude along its depth is not sustainable. To address these challenges, researchers have explored the incorporation of a thin layer to guide the SH wave, and then the wave can decay along with the depth of another layer attached to the thin layer. Such SH waves are called Love surface acoustic waves (L-SAW) [11]. This additional layer effectively confines the energy of acoustic waves to the sensor’s surface, leading to the development of a more sensitive L-SAW sensor [12,13]. Moreover, the waveguiding layer offers the added benefit of protecting the electrodes and piezoelectric materials from corrosion when operating in liquid environments. This makes L-SAW sensors highly valuable for enhancing detection technology in liquid environments. Currently, the effects of various materials such as PMMA, Novolak photoresist, and silicon dioxide (SiO_2_) have been investigated, as potential waveguiding layers on the propagation of SH-SAW [14,15]. Among the waveguiding materials studied, SiO_2_ stands out due to its superior chemical stability and resistance to degradation. Additionally, it has a relatively low shear wave velocity of 2850 m/s, making it an advantageous choice as a waveguiding layer. SiO_2_ can effectively reduce the insertion loss of the acoustic device and enhance the sensitivity of the sensor, making it an attractive option for sensor applications. Zhang et al. [16] fabricated the SH-SAW device on an ST-90° X quartz substrate, deposited a SiO_2_ waveguiding layer with a thickness of 3 μm on its surface using ion-enhanced chemical vapor deposition (PECVD), and completed the multi-channel detection of marine toxins with the device. Wang et al. [17] developed a three-dimensional FEM model for the L-SAW sensor, validating its accuracy using prefabricated equipment, and analyzed its scattering parameters (S-parameters), reflection coefficient parameters, transmission parameters, and phase velocity. To further enhance the sensitivity of L-SAW sensors, researchers explored signal amplification techniques for test samples. Lo et al. [18] applied APTES and glutaraldehyde sensing membranes on the L-SAW sensor to detect the concentration of the cancer-related biomarker antigen epidermal growth factor (EFG). It was found that the optimization of sensitive membranes improved the sensitivity of sensors. In another study, Wang et al. [19] designed an L-SAW sensor to achieve high-sensitivity detection of exosomes. Gold nanoparticles were used to amplify the detection signal during the detection process. Experimental results showed that the detection limit of the device was as low as 1.1 × 10^3^ particles/mL. Furthermore, the L-SAW sensor has found extensive applications in detecting minute biological samples, such as carcinoembryonic antigen [20], prostate-specific membrane antigen [21], and influenza A virus [22]. Currently, most L-SAW sensors use a SiO_2_ layer to confine surface acoustic waves to the surface of the sensor. However, this structure limits the phase velocity of the surface acoustic wave, which cannot break through the limit of the wave velocity of the piezoelectric substrate and SiO_2_. Consequently, this leads to a low shift value of the phase velocity of the acoustic wave, limiting the sensor’s sensitivity when subjected to a load.

When particles or samples with small quantities are applied for detection, the sensitivity of the device significantly influences its detection accuracy, making it crucial to optimize the sensor’s sensitivity. In L-SAW devices, introducing a high wave velocity material with appropriate thickness to the waveguiding layer significantly increases the phase velocity of the acoustic wave and the particle displacement on the device’s surface [23]. This structure can not only confine the acoustic wave energy to the sensor’s surface, providing protection, but also can enhance the sensitivity to changes in the surface environment, thereby improving the overall sensitivity of the sensor. In this paper, a delay line L-SAW sensor with a double-covered waveguiding layer is proposed. The output response, phase velocity of the acoustic wave, and the mass sensitivity were analyzed by FEM. By exploring the structure of the waveguiding layer, it is expected to obtain L-SAW sensors with high sensitivity.

## 2. Device Design and Simulation Methods

### 2.1. Device Design and Working Principle

In this work, a delay line structure was employed for the sensor. A double-covered layer was deposited on the surface of the piezoelectric substrate to serve as the waveguiding layer. The schematic diagram of the device is shown in Figure 1a. Various piezoelectric materials have been utilized to excite L-SAW, with 36°Y-X LiTaO_3_ being particularly noteworthy due to its large wave velocity, significant electromechanical coupling coefficient (K^2^), and relatively low temperature coefficient of frequency (TCF) [24,25,26]. Table 1 displays the parameters of common piezoelectric substrate materials. Moreover, 36°Y-X LiTaO_3_ boasts high consistency and low insertion loss, making it a widely used piezoelectric material in commercial products. Thus, in this work, 36°Y-X LiTaO_3_ was chosen as the material for the piezoelectric substrate. In traditional SAW sensors, copper and gold electrodes with high densities are often used to prevent energy leakage and suppress Rayleigh-mode spurious response. However, this practice results in a phase velocity of SAW that is typically less than 3000 m/s, leading to a significant reduction in sensor sensitivity [19,27]. To mitigate the phase velocity reduction caused by the mass load of the electrodes, Aluminum (Al) was employed as the material for the interdigital electrodes in this study, given its lower density. 

Furthermore, to further enhance the phase velocity of SAW, a layer of Si_3_N_4_ with a high wave velocity of 6040 m/s was deposited on the surface of the piezoelectric substrate. Studies have indicated that the layer with high wave velocity can effectively confine the wave to the layer with lower wave velocity [28]. In this study, a layer of SiO_2_ was deposited on top of the Si_3_N_4_ layer, as illustrated in Figure 1b. The parameters of both SiO_2_ and Si_3_N_4_ materials are listed in Table 2. With this layered structure, the SAW can be better confined to the surface of the sensing region, minimizing energy leakage during the SAW propagation process. Additionally, the SiO_2_ layer addresses the issue of poor stability caused by TCF, thereby contributing to the overall stability of the sensor.

When a sinusoidal voltage signal is applied to the SAW device at a frequency similar to its resonant frequency, resonance occurs on the surface of the piezoelectric substrate, leading to the excitation of L-SAW through the inverse piezoelectric effect. This L-SAW propagates through the waveguiding layer and reaches the output interdigital electrodes. When mass loads are applied to the sensor’s surface, the resonance state of the detection region is altered, causing changes in the phase velocity, SAW amplitude, and resonance frequency of the device. By measuring the frequency shift or phase shift of the SAW oscillation system with a network analyzer, the quantitative detection of changes in the surface of the sensor is realized, such as the changes in micro-mass on the surface.

### 2.2. Simulation Methods and Modelling

FEM analysis of SAW devices can significantly enhance the design accuracy of sensors. Simultaneously, structural optimization can help reduce unnecessary material losses. A FEM model of the L-SAW sensor was developed to analyze the influence of external factors on the resonance state through the coupling of the acoustic field and electric field. The simulation results can serve as valuable references for future designs and optimizations of the L-SAW sensor.

In this work, the delay line L-SAW sensor was simulated using the commercial finite element simulation software COMSOL 5.6 Multiphysics. To account for the polarization of the piezoelectric substrate, an Euler angle (0°, 54°, 0°) was defined to achieve the 36°Y-X orientation of the LiTaO_3_ piezoelectric substrate. In the simulation analysis of the SAW device, the SAW can be treated as the superposition effect of excitation signals from multiple pairs of interdigital transducers (IDTs). However, superimposing excitation signals of all IDTs in the 3D model will complicate the simulation calculation and significantly increase computational costs. The number of IDTs mainly affects the intensity of the excited acoustic wave and is not directly related to the impact of the double-covered waveguiding layer on the sensitivity of the sensor. Studies have shown that simplification can be made according to the periodic characteristics of IDTs [29]. In this paper, only two pairs of electrodes are used for both the input and output IDTs. The schematic diagram of the IDTs is presented in Figure 2a. When the width of the electrode is half of the finger pitch, it is referred to as a uniform IDT, having the highest energy density. To reduce acoustic energy leakage and the impact of Rayleigh clutter, the thickness of the aluminum electrode was set to 0.1 λ [30], where λ represents the wavelength of the SAW, and in this model, λ = 4a. The specific parameters of the simulation model are illustrated in Figure 2b and summarized in Table 3. The simulation model is thoughtfully streamlined, with parameters derived from SAW’s wavelength and propagation properties. To balance computational efficiency, accuracy, and SAW propagation integrity, the delay line area (L_1_) was truncated to 2.5 λ, and the sensing area (L_2_) to 2 λ.

Considering the number of meshes and the computational cost, a segment in the direction of the acoustic aperture was intercepted in this model for analysis. Periodic boundary conditions were used at both ends along the direction of the aperture. To reduce the effect of reflected waves due to the simplification of the model on the propagation of SAW, the ends perpendicular to the direction of the acoustic aperture and the bottom surface of the sensor were set to perfectly matched layers (PML), as shown in Figure 3a. To optimize the calculation results of the simulation analysis, a more regular mapping mesh was utilized in this model. The mesh was generated by sweeping in the direction of the acoustic aperture, and the resulting mesh model is shown in Figure 3b. Research has indicated that when more than six grid nodes are created within each wavelength range in the simulation model, the frequency response becomes independent of the grid size [17]. In this particular model, seven grid nodes were set within each wavelength range along the direction of SAW propagation to mesh the entire sensor model, thereby regulating the mesh length in the acoustic wave propagation direction to 1.5 μm. This meticulous approach ensures both the integrity of the meshes and the precision of computations. Notably, we have taken measures to uphold the aspect ratio of rectangular mesh elements within the range of 6 to 7, further contributing to computational accuracy. It is noteworthy that the simulation model consists of approximately 6500 units.

## 3. Results and Discussion

### 3.1. Simulation Results of Surface Displacements and Phase Velocity

To analyze the vibration state of the sensor during the propagation of L-SAW, a sinusoidal voltage signal was applied to the input IDTs of the FEM model, and the related physical quantity was calculated. The odd fingers of the input and output IDTs were connected and set as the terminal, while the even fingers were connected and grounded. An electrical signal with a power of 2 W was applied to the terminal at the input IDT, while the initial power at the terminal of the output IDT was set to 0. Frequency domain calculation was utilized to analyze the propagation of SAW and the changes in the output signals in the FEM model. Figure 4a illustrates the total surface displacement of the sensor model, which contains a SiO_2_ waveguiding layer with a thickness (h_SiO2_) of 0.2 λ. Figure 4b displays the displacement component of particles on the delay line region when the sensor was in the resonant state. In this model, the displacement component u_y_ of the excited SAW, perpendicular to the propagation direction of SAW on the sensor’s surface, is significantly higher than the components u_x_ and u_z_. These results indicate that the excited SAW in the resonant state achieved by the sensor within the studied frequency range is a horizontal shear wave, confirming the accuracy of the model [24,31].

Assuming that other environmental conditions and the total thickness of the waveguiding layer in the FEM model remained constant, a layer of Si_3_N_4_ with a thickness of 0.1 λ was introduced between the LT piezoelectric substrate and the existing layer of SiO_2_, forming a Si_3_N_4_-SiO_2_ double-covered waveguiding layer. In the FEM model, when the frequency of the input signal resulted in the insertion loss S21 from the input IDT to the output IDT reaching its peak, the sensor was considered to be in a resonant state. The S21 parameter is commonly used to measure and analyze the resonance frequency of a sensor. To reduce computational complexity, analysis, and calculation were focused solely within the resonant frequency range corresponding to the resonant peak when the sensor excited an L-SAW. The resonance frequency of the sensor with the Si_3_N_4_-SiO_2_ double-covered waveguiding layer was compared to a sensor containing only a SiO_2_ waveguiding layer. The results of this comparison are presented in Figure 5.

The results presented in Figure 5 demonstrate that the presence of a Si_3_N_4_ layer with a thickness of 0.1 λ in the waveguiding layer caused an increase in the resonance frequency of the SAW sensor by approximately 11.6 MHz. In the mathematical model, the relationship between the phase velocity and resonance frequency of SAW can be expressed as:(1)$$f_{0}=\frac{v_{s}}{2p}$$

It can be concluded that the presence of the high-wave-velocity Si_3_N_4_ layer increased the phase velocity of the SAW in the sensor, resulting in an increase in the resonance frequency. To investigate further, the total thickness (h_total_) of the waveguiding layer was maintained at 0.3 λ, while the layer thickness ratio of Si_3_N_4_ to SiO_2_ (h_Si3N4_:h_SiO2_) in the double-covered layer structure was varied. Subsequently, the phase velocity of SAW in the sensor was calculated. Figure 6a illustrates the trend of the acoustic phase velocity in the sensor as the layer thickness ratio of Si_3_N_4_ in the waveguiding layer increases.

The results presented in Figure 6a demonstrate that when the total thickness of the waveguiding layer is 0.3 λ, the phase velocity of SAW increases as the thickness of the Si_3_N_4_ layer in the waveguiding layer is increased. Figure 6b depicts the changes in the phase velocity of SAW with varying total thicknesses of the waveguiding layer when the thickness ratio of Si_3_N_4_ and SiO_2_ is different. It is evident that in FEM models with different layer thickness ratios, the phase velocity of SAW decreases as the total thickness of the waveguiding layer increases. Additionally, when the total thickness reaches a certain value, the decrease in phase velocity becomes more significant. This observation can be explained by the behavior of acoustic waves in the waveguiding layers. When the thickness of the SiO_2_ layer is small, its presence reduces the acoustic wave leakage, causing the majority of acoustic waves to exist in the Si_3_N_4_ layer, leading to higher phase velocity values. This finding aligns with previous research [23]. However, as the thickness of the SiO_2_ layer increases, more acoustic waves become confined in the low-wave-velocity waveguiding layer. This results in a more significant decrease in phase velocity until it tends to approach a stable value. Furthermore, when the thickness ratio of the Si_3_N_4_ layer in the waveguiding layer is increased, the downward trend of phase velocity of SAW slows down. This can be attributed to the high wave velocity of the Si_3_N_4_ material, which weakens the limiting effect of SiO_2_ on acoustic waves, thus influencing the phase velocity trend.

### 3.2. Analysis of Surface Average Stress

The sensing mechanism of an L-SAW sensor with a delay line structure is based on the strain generated by mass loads. The strain alters the density, elastic constant, and other parameters of the device surface, leading to a change in the phase velocity of the acoustic wave [32]. Once the initial stress is determined, the SAW propagating in the piezoelectric dielectric can be described by the following equation:(2)$$c_{ijkl}\frac{\partial^{2}u_{l}}{\partial_{x_{j}}\partial_{x_{k}}}+e_{kij}\frac{\partial^{2}\varphi }{\partial_{x_{j}}\partial_{x_{k}}}=\rho \frac{\partial^{2}u_{l}}{\partial t^{2}}$$
(3)$$e_{jkl}\frac{\partial^{2}u_{l}}{\partial_{x_{j}}\partial_{x_{k}}}-\epsilon_{jk}\frac{\partial^{2}\varphi }{\partial_{x_{j}}\partial_{x_{k}}}=0$$

In the given equation, $c_{ijkl}$, $u_{l}$, e, φ, ρ, *t*, and $\epsilon_{jk}$ represent the elastic constants, particle displacements, piezoelectric moduli, electric potential, mass density, time, and dielectric tensors of the material, respectively. The elastic moduli and density of the medium vary with strain. Studies have shown that the mass sensitivity of SAW sensors is related to the stress state and contact stiffness of its surface [33]. In order to explore the influence of Si_3_N_4_ materials on the stress state of the sensor, the stress distribution of the FEM model was analyzed with and without the Si_3_N_4_ layer in the waveguiding layer. The stress distribution of the sensor caused by an external load of 6 × 10^−6^ Pa under the working state was calculated when the total thickness of the waveguiding layer was 0.2 λ, as shown in Figure 7. When the same load is applied, the Si_3_N_4_ waveguiding layer significantly increases the surface stress of the sensor.

To further explore the influence mechanism of the Si_3_N_4_ waveguiding layer on the vibration state of the sensor, the changes in the surface average stress of the sensor were calculated, as shown in Figure 8. Within the calculation range, the surface average stress of sensor models with different ratios of waveguiding layer thickness ramps up and reaches the peak value. When there is no Si_3_N_4_ material in the waveguiding layer, the surface average stress of the sensor reaches the peak value when the total thickness of the waveguiding layer is 0.19 λ. Within a certain range, the surface stress is enhanced with the increase of the thickness ratio of the Si_3_N_4_ layer. Specifically, when the thickness ratio of the Si_3_N_4_ layer to the SiO_2_ layer is 1:2, and the total thickness of the waveguiding layer is 0.26 λ, the surface average stress of the sensor reaches the peak value. The peak value is increased by approximately 45% compared to the sensor model without Si_3_N_4_ material. According to the piezoelectric equation and the description in previous studies, the increase in surface stress and strain corresponds to the increase of the change of phase velocity, which will lead to an increase in the sensor’s sensitivity to mass loads. 

### 3.3. Analysis of Mass Sensitivity

When the L-SAW sensor is utilized for detection in a liquid environment, it is commonly employed for analyzing biological and chemical samples with small volumes and light weights, such as exosomes. Exosomes carry valuable information about their parent cells, and changes in the content of specific exosomes in body fluids are often used as indicators for disease diagnosis. Given the lightweight and low content of exosomes in body fluid samples, the L-SAW sensor plays a critical role in quantitatively detecting exosomes. Effectively identifying small changes in the detection signal is crucial. The unique detection environment demands high sensitivity from the L-SAW sensor to detect minute changes caused by exosomes in the detection region. The sensitivity of a device can be determined by changes in frequency shift, phase shift, and insertion loss per unit load. In this case, the wave propagation process on the piezoelectric substrate involves multiple anisotropic layers, piezoelectric layers, double-covered waveguiding layers, and three-dimensional wave diffraction. As such, accurately predicting the sensitivity of SAW devices under real operating conditions becomes complex. Analyzing all the characteristics of acoustic waves in a single method is challenging. Given that the L-SAW sensor’s sensitivity is primarily affected by the stress and strain induced by the mass loads of the sample, this study focuses on investigating the influence of the double-covered waveguiding layer on the mass sensitivity of the sensor. The mass sensitivity can be expressed as:(4)$$S_{m}=\frac{dv}{v\cdot dm}=\frac{df}{f\cdot dm}=\underset{∆m\to 0}{\lim}\frac{∆f}{f\cdot ∆m}$$

In the multi-thin layer system of this scheme, the mass sensitivity of the sensor will vary with the total thickness of the waveguiding layer and the layer thickness ratio of SiO_2_ and Si_3_N_4_. This work explored the influence mechanism of waveguiding layer structure on the mass sensitivity of the sensor, and the sensor structure with higher sensitivity was expected. 

In this work, the quantitative detection of exosomes in body fluid samples was used as an example to analyze the mass sensitivity of the sensor. Exosomes, which have a diameter of 30–150 nm, were approximated as spheres with a diameter of 100 nm and a material density of 1150 kg/m^3^ [34]. The concentration of exosome samples typically used in experiments is around 10^10^ particles/mL [19]. For our sensor design, the surface area of the sensing region is 1 cm^2^. Due to the extremely small size of exosomes, it is thought that when they are captured on the surface of the sensing region, they only affect the quality of the surface. In this model, the change in mass density of the sensing region’s surface was used to simulate the exosome capture process. Assuming that 0.01 mL of exosome sample is applied on the surface to the sensing region for each measurement, resulting in a mass density of about 6 × 10^−7^ kg/m^2^. The resonance frequency of the sensor model was calculated by analyzing the S21 parameters both without loads and with loads. The total thickness of the waveguiding layer of the sensor is 0.2 λ which contains only a SiO_2_ layer. The simulation results indicate that a simulated load of 6 × 10^−7^ kg/m^2^ on the sensing region results in a frequency shift of about 8.5 × 10^−3^ MHz compared to the vibration state without mass load.

As can be seen from Figure 9a, the shift of resonant frequency of the sensor exhibits a linear relationship with the mass load, and the mass sensitivity *S_m_* of the sensor can be calculated by analyzing its slope. To investigate the influence of the Si_3_N_4_ layer on the sensitivity of the sensor, the thickness of the SiO_2_ layer was maintained at 0.2 λ, while different thicknesses of the Si_3_N_4_ layer were added to the waveguiding layer. The variation in sensitivity with the thickness of the Si_3_N_4_ layer is illustrated in Figure 9b. When the thickness of the Si_3_N_4_ layer (h_Si3N4_) is 0.05 λ, the sensitivity of the sensor reaches a peak value. Compared to sensors with only a SiO_2_ waveguiding layer, the mass sensitivity is improved by approximately 48.3%. Furthermore, it can be observed that within a certain range, the sensitivity increases with the augmentation of the Si_3_N_4_ layer’s thickness. This phenomenon occurs because, as the proportion of Si_3_N_4_ increases, the phase velocity of SAW also increases, leading to a higher change in phase velocity caused by a unit mass load.

The influence of the total thickness of the waveguiding layer on the sensitivity of the sensor was also explored. It was observed that within a certain range, the mass sensitivity of the sensor increases and reaches a peak as the total thickness of the waveguiding layer increases, as shown in Figure 10a. The peak value of mass sensitivity represents the sensitivity of the sensor with a specific layer thickness ratio. Furthermore, to delve deeper into the impact of the Si_3_N_4_ layer on the mass sensitivity of the sensor, a comparison was made between the peak mass sensitivity values of sensors with different layer thickness ratios, as shown in Figure 10b. When the thickness ratio of the Si_3_N_4_ layer to the SiO_2_ layer is 1:2, and the total thickness of the waveguiding layer is 0.26 λ, the mass sensitivity of the sensor is measured to be 63.55 m^2^/kg. Compared to the traditional sensor model with only a SiO_2_ waveguiding layer, the peak mass sensitivity has been improved by approximately 50%. The change in mass sensitivity is related to the change in surface average stress at the delay line region when the layer thickness ratio is different. Therefore, the increase in mass sensitivity is attributed to the increase of surface average stress at the delay line region due to the inclusion of Si_3_N_4_ layers in the waveguiding layer.

## 4. Conclusions

In this paper, an L-SAW sensor with a double-covered waveguiding layer was proposed and a comprehensive analysis of the device was conducted in the frequency domain using FEM. The incorporation of Si_3_N_4_ material with high wave velocity in the waveguiding layer effectively enhances the phase velocity of L-SAW as it propagates through the waveguiding layer. Further analysis of the mass sensitivity of the SAW device reveals that the Si_3_N_4_ layer also contributes to an improved mass sensitivity of the sensor. Within a specific range, when the thickness ratio of the Si_3_N_4_ layer to the SiO_2_ layer is 1:2, and the total thickness of the waveguiding layer is 0.26 λ, the mass sensitivity of the sensor reaches 63.55 m^2^/kg. This represents an impressive improvement of approximately 50% compared to the mass sensitivity of the traditional sensor model equipped with only a SiO_2_ waveguiding layer. The increase in mass sensitivity can be attributed to the increase in phase velocity of SAW and surface average stress of the delay line region due to the Si_3_N_4_ layer in the waveguiding layer. The simulation results presented in this paper serve as valuable guidance for the design and development of subsequent high-sensitivity SAW sensors.

## Figures and Tables

**Figure 1 micromachines-14-01696-f001:**
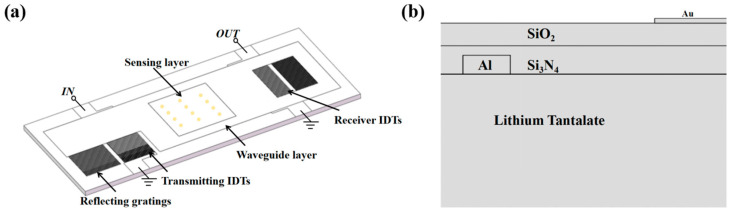
(**a**) Schematic diagram of the designed L-SAW device structure. (**b**) Conceptual view of the sensor with Si_3_N_4_-SiO_2_ double-covered waveguiding layer.

**Figure 2 micromachines-14-01696-f002:**
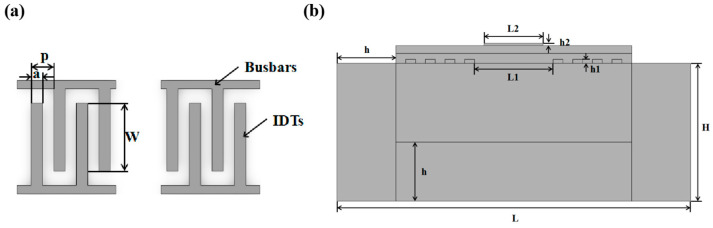
(**a**) Schematic diagram of the IDT. (**b**) Geometric model for FEM simulation.

**Figure 3 micromachines-14-01696-f003:**
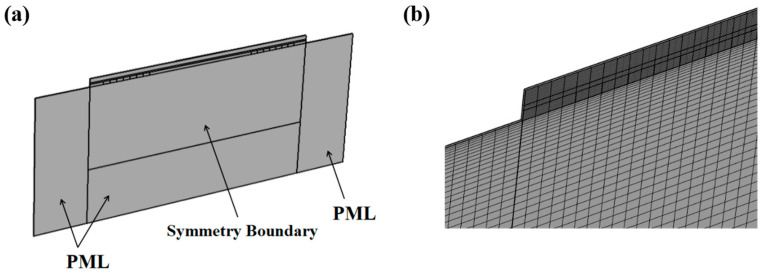
(**a**) The 3D model of the device used in COMSOL. (**b**) The partial mesh of the model.

**Figure 4 micromachines-14-01696-f004:**
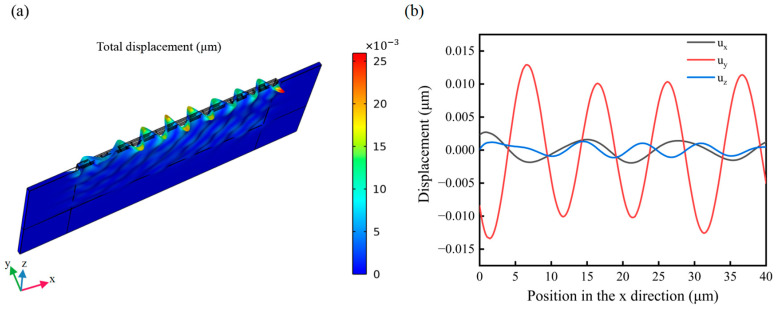
(**a**) Simulation diagram of the total surface displacement of the model. (**b**) The plot of displacements u_x_, u_y_, and u_z_ at the surface of the sensor.

**Figure 5 micromachines-14-01696-f005:**
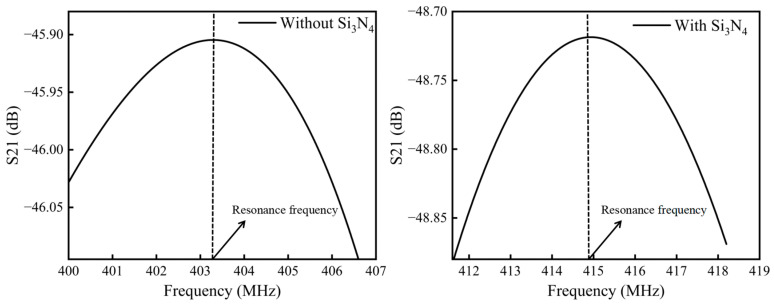
Insertion loss at different frequencies of L-SAW devices with and without Si_3_N_4_ waveguiding layer.

**Figure 6 micromachines-14-01696-f006:**
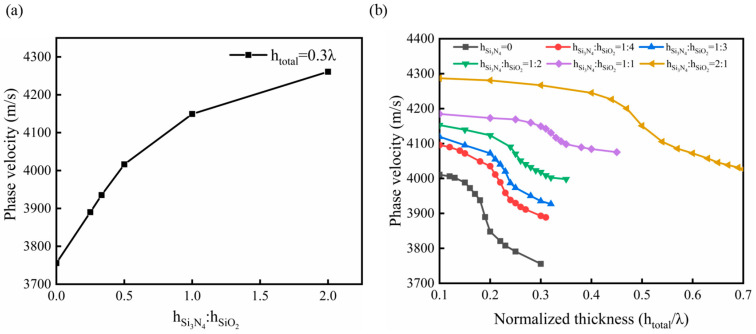
Relationship between phase velocity of SAW and (**a**) the ratio of Si_3_N_4_ layer thickness and (**b**) thickness of waveguide layer with different ratios of Si_3_N_4_ and SiO_2_.

**Figure 7 micromachines-14-01696-f007:**
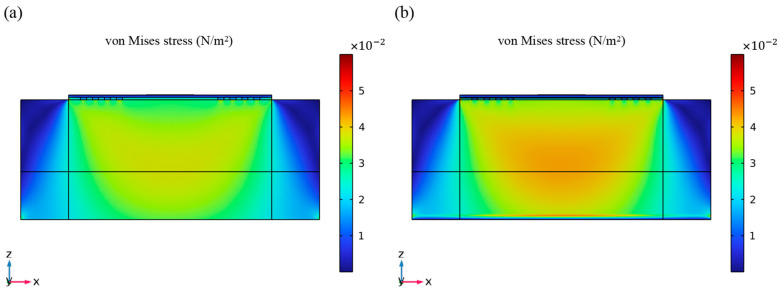
The stress contour of the model after applying load (**a**) without the Si_3_N_4_ layer and (**b**) with the Si_3_N_4_ layer.

**Figure 8 micromachines-14-01696-f008:**
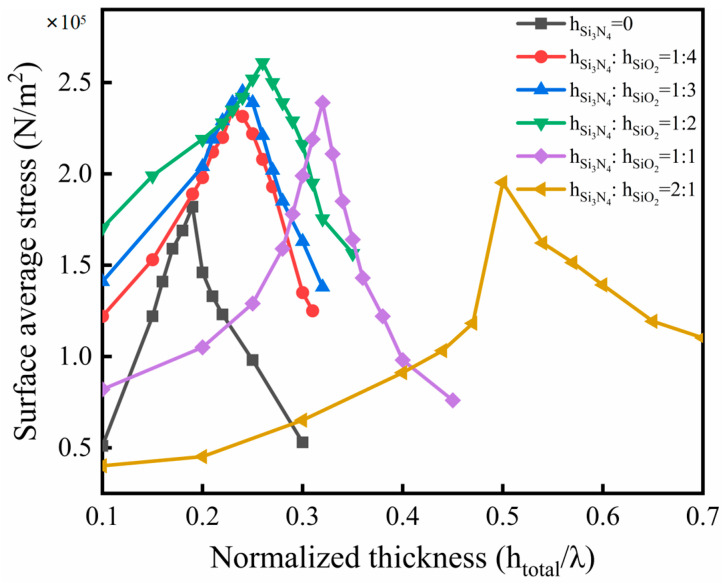
Relationship between the surface average stress of the sensor and the total thickness of the waveguiding layer.

**Figure 9 micromachines-14-01696-f009:**
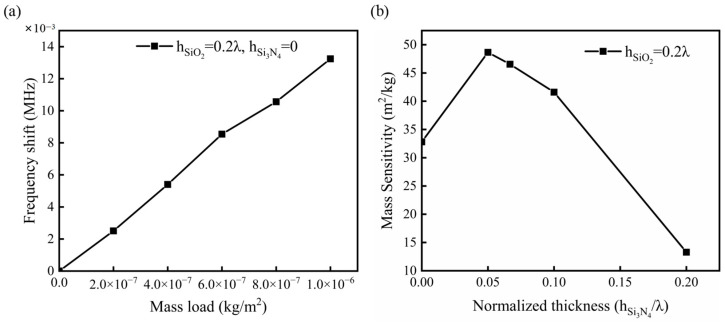
(**a**) Relationship between frequency shift of the sensor and mass load; (**b**) Relationship between mass sensitivity of the device and the layer thickness of Si_3_N_4._

**Figure 10 micromachines-14-01696-f010:**
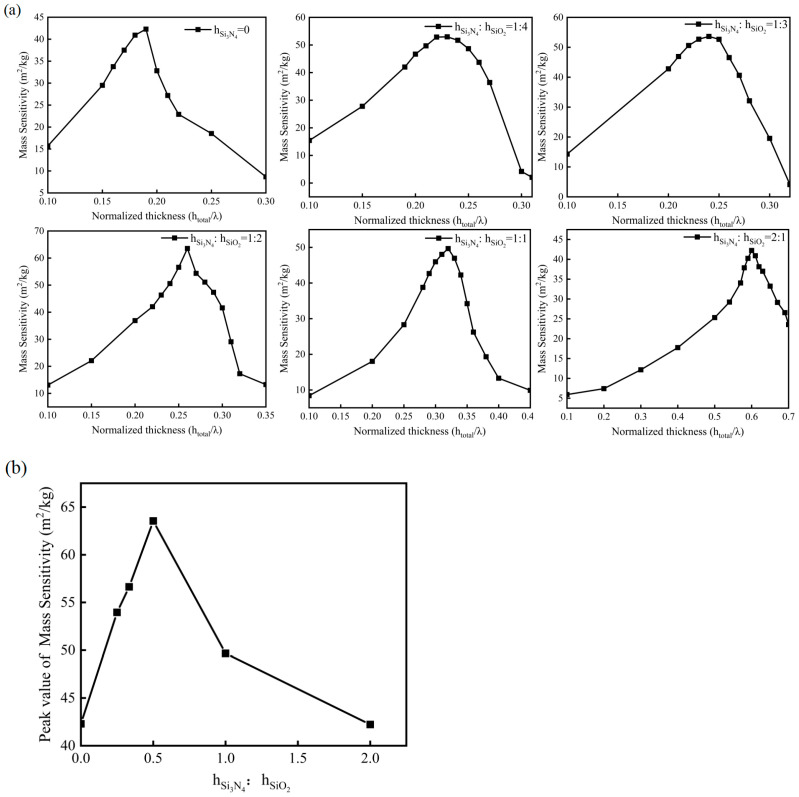
(**a**) Relationship between the mass sensitivity and the total thickness. (**b**) Relationship between the peak value of mass sensitivity and layer thickness ratio.

**Table 1 micromachines-14-01696-t001:** Parameters of common piezoelectric materials.

Piezoelectric Materials	Wave Velocity (m/s)	Electromechanical Coupling Coefficient *K*^2^ (%)	Temperature Coefficient of Frequency (10^−6^/°C)
ST-90° Quartz	5050	0.1	0
36°Y-X LiTaO_3_	4200	5	−32
36°Y-X LiNbO_3_	4690	10.5	−70
128°Y-X LiNbO_3_	3980	5.5	−74

**Table 2 micromachines-14-01696-t002:** Parameters of SiO_2_ and Si_3_N_4_ material.

Waveguiding Materials	SiO_2_	Si_3_N_4_
Density (kg/m^3^)	2200	3100
Phase velocity (m/s)	2850	6040
Relative permittivity	4.2	9.7
Young’s modulus (Pa)	70 × 10^9^	250 × 10^9^
Poisson’s ratio	0.17	0.23

**Table 3 micromachines-14-01696-t003:** Simulation parameters and device dimensions.

Substrate height, H	5 λ	Sensing layer length, L_2_	2 λ
Substrate length, L	10.5 λ	IDT height, h_1_	0.1 λ
Substrate width, T	0.2 μm	Sensing layer height, h_2_	0.1 μm
PML width, h	2 λ	Electrode width, a	0.25 λ
Delay path length, L_1_	2.5 λ	Finger pitch, p	0.5 λ

## Data Availability

Not applicable.

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
