# Peer review of "Finite Element Study for Mass Sensitivity of Love Surface Acoustic Wave Sensor with Si3N4-SiO2 Double-Covered Waveguiding Layer"

_micromachines, 2023, doi:10.3390/mi14091696_

Round 1
Reviewer 1 Report
1. How are the simulation parameters determined in table 3.
2. For a delay line, not only the phase velocity but also the insertion loss (Q factor) will affect the sensitivity, the insertion loss of the delay line with Si3N4 is higher as shown in fig.5. This problem should be considered.
Reviewer 2 Report
The article proposed a novel surface acoustic wave sensor with Si3N4-SiO2 double-covered waveguiding layer and studied the finite element for mass sensitivity. The article can be considered for publication in "Micromachines" after revising the following questions. The comments are below.
1) Line 12 “a L-SAW” will be “an L-SAW”, Line 42 “substrate” will be “substrates” etc. Do check the manuscript again for grammatical errors and typos. Please ask an English speaker colleague to do a proof read or use the journals English assist feature to polish the work.
2) In section 2.1, the author mentioned that Aluminum (Al) was employed as the material for the interdigital electrodes in this study, given its lower density. In view of a variety of materials are used in the sensor, the location of metallic aluminum should be shown in the figure.
3) In section 2.2, the author proposed that two pairs of electrodes are used for both the input and output IDTs. Did the authors consider the effect of more electrodes on the simulation?
4) In section 2.2, the author divided the model in the form of meshes. It can be seen that the shape of meshes is rectangle. I recommend illustrating the size and number of meshes to help to understand the finite element.
5) Are the simulation results of surface displacements and phase velocity adequately validated with experimental data or other independent methods?
6) In section 2.2, the author proposed a principle called PML. Compared to other layers, what the difference of PML?
7) In section 2.2, the author proposed that research has indicated that when more than six grid nodes are created within each wavelength range in the simulation model, the frequency response becomes independent of the grid size. Is there a theoretical basis for the research here? If yes, Please cite relevant literature.
8) Is the introduction of the Si3N4-SiO2 double-covered waveguiding layer practically feasible for the real-world implementation of the sensor? Are there any potential manufacturing challenges or cost considerations?
9) How generalizable are the findings of this study to other types of sensors or materials? Are there specific applications or scenarios where the Si3N4-SiO2 waveguiding layer might not provide the same enhancements in sensitivity or performance?
10) In section 3.1, the ux, uy and uz can be written as a superscript such as ux, uy, uz.
Reviewer 3 Report
In this paper, an L-SAW sensor with a thin Si3N4-SiO2 double-covered layer was proposed for samples with small mass loads. The output response, phase velocity of the acoustic wave, and the mass sensitivity were analyzed using the finite element method (FEM). The simulation results show that the Si3N4 layer with high wave velocity greatly weakens the limitation of SiO2 on the phase velocity. Within a certain range, the mass sensitivity of the sensor is enhanced with the increase in the total thickness of the waveguiding layer and the thickness ratio of Si3N4 in the double-covered layer. The surface average stress of the delay line region follows the same trend as the mass sensitivity. This L-SAW sensor, featuring a double-covered waveguiding layer, demonstrates high sensitivity and a simple structure. The manuscript has a reference value for the design of SAW sensors. But the followed questions and suggestions should be answered before acceptation.
(1) In line 62, “This additional layer effectively confines the energy of acoustic waves to the sensor's surface, leading to the development of a more sensitive Love surface acoustic wave (L-SAW) sensor[11,12].”, before doing so, the author should briefly describe what Love surface acoustic wave.
(2) There are some writing format problems in the manuscript, such as, in line 68, there is an extra space between “Currently,” and “the”. In line 76, “3μm” should be written as “3 μm”, and so on. The writer should re-examine the grammar and formatting of the article.
(3) In line 275, “6×10-6 Pa” should be written as “6×10-6 Pa”.
(4) In line 268, Thes symbols c, u, e, Ï•, ρ, t and εdoesn't correspond to the equation (2) and equation (3). I strongly encourage authors to check the details in the manuscript carefully before resubmitting it.
(5) All the pictures in the article seem to be not clear, it is recommended to re-modify the resolution of these pictures.
(6) Within a specific range, when the thickness ratio of the Si3N4 layer to the SiO2 layer is 1:2, and the total thickness of the waveguiding layer is 0.26λ, the mass sensitivity of the sensor reaches 63.55m2/kg. Why thickness ratio of the Si3N4 layer to the SiO2 layer is 1:2, and why the total thickness of the waveguiding layer is 0.26λ? If it's 1:3, 0.3λ, what's going to happen? Is there any basis for that?
Minor editing of English language required
Round 2
Reviewer 2 Report
The author has revised the manuscript well. I suggests publication.